

# Riverine inputs and phytoplankton community composition control nitrate cycling in a coastal lagoon

Mindaugas Zilius[1*], Rūta Barisevičiūtė[2], Stefano Bonaglia[1,3], Isabell Klawonn[4], Elise Lorre[1], Tobia Politi[1,3], Irma Vybernaite-Lubiene[1], Maren Voss[4], and Paul A. Bukaveckas[5]

[1]Marine Research Institute, Klaipeda University, Klaipeda, 92294, Lithuania
[2]State Research Institute, Center for Physical Sciences and Technology, 02300 Vilnius, Lithuania
[3]Department of Marine Sciences, University of Gothenburg, Box 461, 405 30 Gothenburg, Sweden
[4]Department of Biological Oceanography, Leibniz Institute for Baltic Sea Research, 18119 Rostock, Germany
[5]Center for Environmental Studies, Virginia Commonwealth University, Richmond, VA 23284, USA

*Correspondence to*: Mindaugas Zilius (mindaugas.zilius@jmtc.ku.lt)

**Abstract.** Estuarine systems, being situated at the interface between land and marine environments, are important sites for nitrate ($NO_3^-$) retention and processing due to large inputs, long retention time, and high biogeochemical activity. However, it remains uncertain how pelagic and benthic processes control $NO_3^-$ cycling and how these differ between contrasting seasons. In this study, we measured pelagic and benthic assimilatory and dissimilatory $NO_3^-$ processes in a large lagoon (Curonian Lagoon, SE Baltic Sea) to understand changes in $NO_3^-$ cycling in relation to variation in riverine inputs and shifts in phytoplankton community composition. We show that in spring, benthic dissimilatory and assimilatory $NO_3^-$ processes were important, while in summer, pelagic assimilatory processes dominated. During spring, diatom blooms promote greater delivery of nitrogen (N) and labile organic matter to the benthos resulting in greater denitrification in the sediments and a net flux of $NO_3^-$ from the water column to the sediments. In summer, phytoplankton blooms dominated by buoyant cyanobacteria exhibited high rates of assimilatory uptake and greater particulate organic N export to the sea, but low rates of sediment–water exchange. Cyanobacteria blooms were associated with higher absolute rates of $NO_3^-$ uptake, as well as higher mass-specific rates compared to spring. Given the low dissolved inorganic N in summer, high uptake indicates that the pelagic community possessed a nutritional strategy to efficiently utilize multiple N forms. Overall, our findings show that the seasonal succession from diatom to cyanobacteria-dominated communities is associated with a shift from strong benthic-pelagic coupling to predominantly pelagic-based N cycling.



# 1 Introduction

Nitrate ($NO_3^-$) frequently constitutes most of the riverine nitrogen (N) load from land to coastal systems worldwide (Peierls et al., 1991; Vybernaite-Lubiene et al., 2018). Agriculture is a major source of $NO_3^-$ resulting in high concentrations in rivers and groundwater, where catchments are dominated by row crop farming (Arheimer et al., 2012; Santos et al., 2021). While implementation of strategies to mitigate N loss from farmland has progressed in recent decades, $NO_3^-$ loads in some regions continue to increase (e.g. Vybernaite-Lubiene et al., 2018). Estuarine systems, being situated at the interface between land and seas or oceans, are potentially important sites for $NO_3^-$ retention due to large inputs, long retention time, and high biogeochemical transformation rates (Voss et al., 2011; Asmala et al., 2017).

The attenuation of N through-puts in lagoons and estuaries is referred to as the "filter" function. This important ecosystem service is conventionally linked to processes occurring in sediments (Voss et al., 2010; Anderson et al., 2014; Carstensen et al., 2020; Magri et al., 2021). Here, denitrification is the main microbial process that removes $NO_3^-$, since anammox is typically low in estuarine systems (Thamdrup, 2012). $NO_3^-$ can be also transformed by dissimilative nitrate reduction to ammonium (DNRA), but in contrast to denitrification, the recycled N remains within the system (Giblin et al., 2013; Magri et al., 2021). In the water column, phytoplankton and bacteria uptake is an important mechanism for converting $NO_3^-$ to particulate and dissolved organic N forms (PON and DON; Middelburg and Nieuwenhuize, 2000a; Olofsson et al., 2019). After incorporation into biomass, particulate may N settle onto surface sediments where it is mineralized or buried (Brion et al., 2008).

Various factors can impact $NO_3^-$ assimilation and conversion to other forms. For example, seasonally blooming phytoplankton species differ in their nutrient uptake strategies (Dortch, 1990; Middelburg and Nieuwenhuize, 2000a; Berg et al., 2003; Lomas and Glibert, 2003). Higher concentrations of $NO_3^-$ typically lead to increased uptake rates of phytoplankton and bacteria, which in turn leads to an increase in biomass (Middelburg and Nieuwenhuize, 2000b; Twomey et al., 2005; Glibert et al., 2015). Elevated levels of $NO_3^-$ also stimulate the rates of denitrification or DNRA in sediments (Dong et al., 2000, 2009; Magri et al., 2021). The balance between these two dissimilatory processes may be influenced by the amount of organic carbon in the sediment, the presence of reduced compounds (e.g. $H_2S$, $Fe^{2+}$) and the availability of $NO_3^-$ (Kessler et al., 2018; Murphy et al., 2020). Where $NO_3^-$ respiring bacteria in photic sediments are exposed to light, they compete for $NO_3^-$ with phytoplankton or microphytobenthos which could lead to a suppression of their activity (Sundbäck et al., 2006; Risgaard-Petersen et al., 2005; Bartoli et al., 2021).

Overall, the utilization of $NO_3^-$ is contingent upon its presence in the water column, which is subject to regulation by riverine inputs, particularly in temperate and boreal estuaries with large seasonal variations in river flow (Bukaveckas et al., 2018; Zilius et al., 2018). This, in turn, can affect the rate processes that transform $NO_3^-$ (Veuger et al., 2004; Killberg-Thoreson et al., 2020). Therefore, it is reasonable to expect that there are different mechanisms at play in the cycling of $NO_3^-$ throughout the year. Concurrent measurements that account for these temporal variations would improve our understanding of seasonal



variations in the supply and demand for N. Although $NO_3^-$ cycling is of considerable importance to understanding eutrophication and recovery, few studies have quantified the multiple processes responsible for $NO_3^-$ retention and removal (i.e. denitrification, DNRA, and $NO_3^-$ production) in coastal systems (e.g. Bartl et al., 2019; Broman et al., 2021). Simultaneous

measurements of pelagic assimilatory processes, such as $NO_3^-$ uptake by phytoplankton and bacteria in the water column, and benthic dissimilatory processes, such as denitrification, anammox and DNRA, would provide insight into how N is transformed within the estuarine systems.

In the present study, we measured assimilatory and dissimilatory $NO_3^-$ processes in both the sediment and water column of a large oligohaline lagoon (Curonian Lagoon, SE Baltic Sea). The objectives were (1) to describe the seasonal dynamics of $NO_3^-$

cycling in the context of variable riverine inputs and changes in phytoplankton community composition, and (2) to assess the importance of $NO_3^-$ cycling in the context of N through-puts to the sea. This study builds on our earlier work documenting nutrient mass balances in the Curonian Lagoon (Vybernaite-Lubiene et al., 2017, 2022; Zilius et al., 2018) by quantifying specific mechanisms of N assimilation and conversion. We hypothesize that during typical spring conditions, when diatoms dominate and riverine inputs are elevated, N cycling is driven by strong benthic–pelagic coupling as settling diatoms deliver

N to the benthos and high $NO_3^-$ concentrations in the lagoon favor diffusive flux into the sediments. During typical summer conditions, cyanobacteria dominate and lagoon $NO_3^-$ concentrations are low. Under these circumstances, N cycling will be dominated by pelagic processes as the high biomass of positively buoyant cyanobacteria drives rapid uptake and re-mineralization in the water column. Overall, by measuring the complex biogeochemical interactions that shape nutrient dynamics in the lagoon systems, we can develop a more robust understanding to inform management and policy decisions

aimed at maintaining ecological health.

## 2 Material and Methods

### 2.1 Study site

The Curonian Lagoon is a large (1584 $km^2$), shallow (mean depth 3.8 m) waterbody located along the southeast coast of the Baltic Sea (Fig. 1). The lagoon is mainly freshwater (mean salinity = 0.2 PSU) due to large riverine inputs and limited exchange

with the Baltic Sea through a narrow channel (Zemlys et al., 2013). The Nemunas River is the principal tributary (16 $km^3$ $yr^{-1}$, Vybernaite-Lubiene et al., 2018) accounting for 96% of total water inputs and the main source of nutrients (Jakimavičius and Kriaučiūnienė, 2013; Vybernaite-Lubiene et al., 2022). The inflow of the Nemunas divides the lagoon into northern (greater riverine influence) and central-southern (more lacustrine) areas that differ in their prevailing biogeochemical processes (Zilius et al., 2014, 2018; Umgiesser et al., 2016). The northern area is characterized by shallower depths (1.5–2 m), shorter

water residence time (seasonal range = 50–100 days), and sandy sediments with low organic matter content ($C_{org}$ < 0.5%) (Umgiesser et al., 2016; Zilius et al., 2018;). The central-southern area of the lagoon is deeper (mean = 3.5 m), has a longer water residence time (seasonal range = 100–250 days), and organic-rich deposits (predominantly silty sediments, $C_{org}$ = 10–14%). The lagoon is vertically well-mixed owing to the shallow depth and weak salinity gradients (Zilius et al., 2014, 2020).



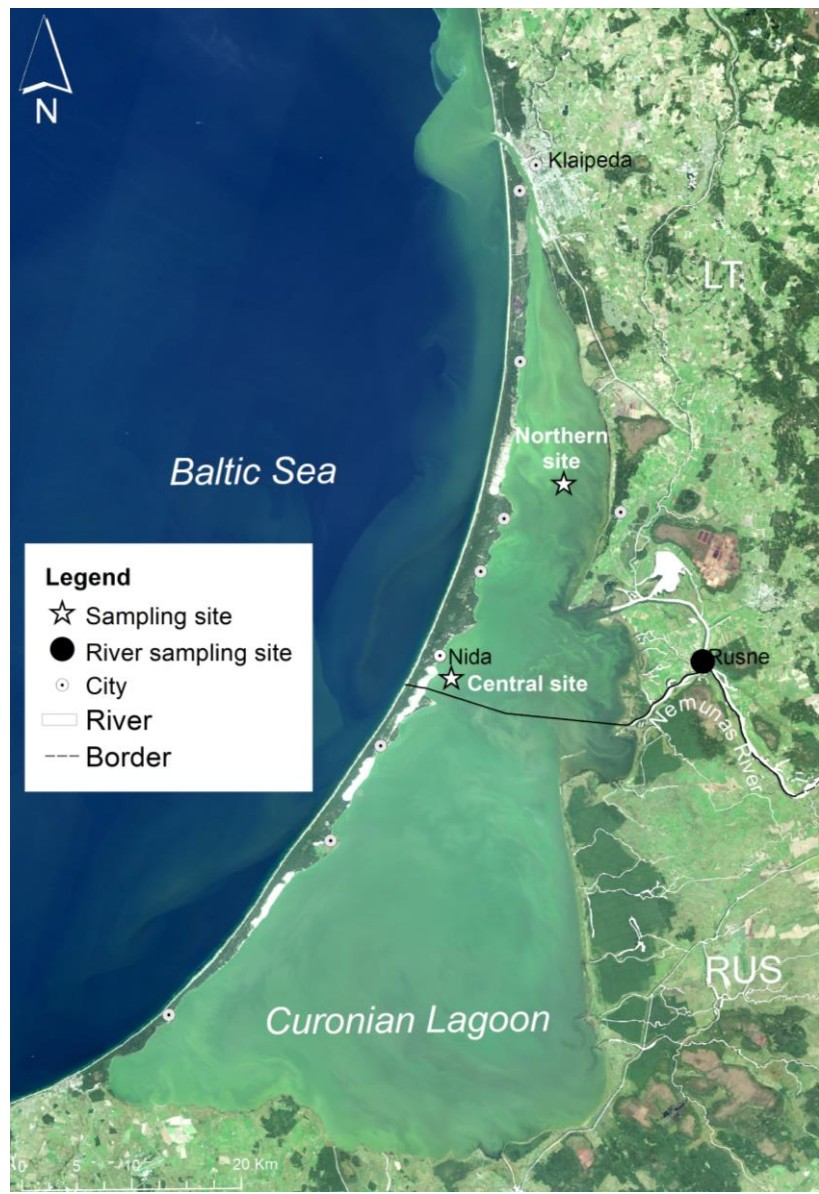

**Figure 1. Satellite image by OLI/Landsat-8 (18/09/2014) showing summer blooms in the Curonian Lagoon with the sampling sites (red circles) representing the northern and south-central regions and monitoring site at the Nemunas River (blue circle). The black line indicates a state border between two countries.**

In this study, internal microbial processes that play a role in the turnover of $NO_3^-$ were measured in the spring (17 May) and summer (23 August) of 2021 at two sites (northern and central areas) in the Curonian Lagoon (Fig. 1) (Zilius et al., 2014, 2021). Assuming shallow depths and a well-mixed water column, vertically integrated water samples that represent the whole water column were collected with a Niskin bottle and transferred to (1) opaque HDPE bottles (2 L) for dissolved nutrient and





particulate matter analyses and (2) 20 L jars for assimilation measurements. In addition, 5 replicate intact sediment cores (i.d. 8 cm, height 30 cm) were collected within 50–150 m of each sampling station using a hand corer for measurement of dissimilatory $NO_3^-$ processes. 150 L of bottom water were also collected for core maintenance during transportation and

incubation activities. During each sampling campaign, vertical profiles of water temperature, salinity, dissolved oxygen, and photosynthetically active radiation (PAR) were measured *in situ* with a YSI ProQuatro multiple probe (YSI Inc.) and an LI-192 underwater quantum sensor (LI-COR Biosciences). We also monitored dissolved inorganic nitrogen (DIN) concentrations in the Nemunas River (Fig. 1) to estimate riverine inputs. River water samples were collected at 2-week intervals from January to December 2021, and monthly during the low flow period (May–September).

**2.2 Water column measurements**

We used stable isotopes of carbon (C) and N to measure net C-fixation and $NO_3^-$ uptake rates based on a method previously described by Montoya et al. (1996), and Dugdale and Wilkerson (1986). Briefly, 20×250 mL polycarbonate bottles were filled with *in situ* water collected from each site (with no head space), sealed, and assigned to light and dark incubations (see a schematic representation in Appendix, Fig. S1). A $H^{13}CO_3^-$ (NaH$^{13}$CO$_3$, 98 atom % $^{13}$C, Sigma Aldrich) tracer was injected to

a final concentration of 0.2 mM. Afterwards, a $^{15}NO_3^-$ tracer (Na$^{15}$NO$_3$, 98 atom % $^{15}$N, Sigma Aldrich) was injected to a final concentration of 50 µM (spring) and 0.5 µM (summer), based on ambient concentrations. Short incubations lasting from 1.5 to 4 h with three time points $T_0$, $T_1$, and $T_2$ were performed in triplicates. Additionally, an unlabeled control bottle was included for each time point. Water samples were placed in an outdoor tank filled with water to maintain temperature (15.0 °C and 21.0 °C, respectively in May and August) and shaded to prevent high light exposure. Samples representing aphotic conditions were

covered with aluminium foil. At each time point, bottles were opened and 70–100 mL of water was filtered on a pre-combusted 25 mm Advantec GF75 glass fiber filter (nominal pore size 0.3 µm) for PO$^{13}$C and PO$^{15}$N analyses. Additional aliquots were collected and (1) transferred without headspace into 12 ml exetainer (Labco) with 100 µl of 7 M ZnCl$_2$ for $^{13}$C in dissolved inorganic carbon (DIC) and (2) filtered and transferred into PE test tubes for nitrate ($^{14}NO_3^-$ + $^{15}NO_3^-$) analysis. All samples were frozen at –20°C until analysis (except for DI$^{13}$C stored at 4°C).

Net $NO_3^-$ uptake and C-fixation, which include phytoplankton and bacterial assimilation into biomass, were calculated following Dugdale and Wilkerson (1986) and Montoya et al. (1996), respectively. Net C-fixation in light bottles was used to estimate the total photosynthetic N demand (inclusive of all forms of N) in the euphotic zone based on the C:N ratio of seston (measured at the initial time point). By combining these two methods, we were able to estimate both the total autotrophic N demand (from C-fixation) and the combined autotrophic and heterotrophic $NO_3^-$ demand. To derive daily values, hourly $NO_3^-$

uptake and C fixation rates in light were used to estimate day-time rates within the euphotic layer taking into account the proportional light dosages during the incubation relative to daily *in situ* values (Bukaveckas et al., 2011). The hourly uptake rates in the dark were used to estimate night-time values and rates in the aphotic layer. The total uptake rates across the water column were calculated from the sum of the depth-integrated uptake in both the aphotic and photic layers.





## 2.3 Benthic $NO_3^-$ flux and process measurements

In the laboratory, open sediment cores were placed into an incubation tank containing unfiltered, aerated and well-stirred estuarine water in a temperature-controlled room (14.5 °C and 21.0 °C, respectively in May and August). A stirring bar, driven by an external magnet at 40 rpm, was inserted in each core approximately 15 cm above the sediment interface to maintain water column mixing while avoiding sediment resuspension (Zilius et al., 2018, 2021). After an overnight preincubation, a gas-tight lid was placed on the top of each core at the start of the dark incubations. For sediment cores collected from the shallower (northern) site in spring, we also conducted incubations under *in situ* light conditions (~60 µmol $s^{-1}$ $m^{-2}$) to evaluate the impact of light on $NO_3^-$ fluxes. All incubations lasted from 3 to 9 h to keep the final oxygen concentration within 20% of the initial value. At the beginning and end of the incubation, 20 mL water aliquots were collected from each core, and filtered (Frisenette GF/F filters) into 12 mL plastic test tubes for later $NO_3^-$ analysis.

After the flux measurements, cores were opened and left submerged in the incubation tank for ~5 h. Afterwards, $NO_3^-$ reduction processes were measured following the unrevised isotope-pairing technique (IPT, Nielsen, 1992), which was appropriate here due to relatively low anammox rates in the lagoon sediments (< 4% of total $N_2$ production; Zilius, 2011). Briefly, all cores were spiked with $^{15}NO_3^-$ tracer (20 mM $Na^{15}NO_3$, 98 atom % $^{15}N$, Sigma Aldrich) to a final $^{15}N$-label percentage between 51% and 100% depending on background concentrations. The cores were then capped and incubated in the dark as described for fluxes. At the end of incubations, the water and the whole sediment were gently mixed to a slurry. Thereafter, 20 mL aliquots of the slurry were transferred into 12 mL exetainers (Labco) allowing twice overflow, and fixed with 200 µL of 7 M $ZnCl_2$ for later $^{29}N_2$ and $^{30}N_2$ analyses. An additional 40 mL subsample was collected and treated with 2 g of KCl for the determination of the exchangeable $NH_4^+$ pool and the $^{15}NH_4^+$ fraction.

Net daily rates were derived by multiplying the hourly rates by the length of the day. When cores were incubated in both light and dark conditions (spring, northern site), the net daily rates were calculated by multiplying hourly rates by the mean length of the light and dark periods.

## 2.4 Analytical and other methods

In the laboratory, water samples were filtered (GF/F filters) within 1–2 h of collection, and transferred into 10 ml PE tubes for dissolved inorganic N analysis (DIN). The concentration of DIN ($NH_4^+$, $NO_2^-$ and $NO_{2+3}^-$) was measured with a continuous flow analyzer (San$^{++}$, Skalar) using standard colourimetric methods (Grasshoff et al., 1983). $NO_3^-$ was calculated by subtracting nitrites ($NO_2^-$) from the combined nitrite and nitrate concentration ($NO_{2+3}^-$). Total dissolved nitrogen (TDN) was analyzed by the high temperature (680 ºC) combustion, catalytic oxidation/NDIR method using a Shimadzu TOC-V 5000 analyzer with a TNM-1 module. Dissolved organic nitrogen (DON) was calculated as a difference between TDN and DIN. Samples for phytoplankton counting were immediately preserved with acetic Lugol's solution and examined at magnifications of 200× and 400× using a LEICA DMI 3000 (Leica Microsystems) inverted microscope. Phytoplankton community





composition was determined using the Utermöhl method (Utermöhl, 1958) according to HELCOM recommendations (HELCOM, 2017). Phytoplankton biomass (mg L$^{-1}$) was calculated according to Olenina et al. (2006).

Filters for PO$^{15}$N and PO$^{13}$C analyses were analyzed with a continuous-flow isotope ratio mass spectrometer (IRMS; Thermo-Finnigan, Delta S, Thermo Fisher Scientific) at the Leibniz Institute for Baltic Sea Research Warnemünde (IOW). Isotopic samples for $^{29}$N$_2$ and $^{30}$N$_2$ production were analyzed by gas chromatography-isotopic ratio mass spectrometry (Thermo Delta

V Plus, Thermo Fisher Scientific) with means of a Conflo III interface at the University of Southern Denmark. Samples for $^{15}$NH$_4^+$ production were analyzed by the same technique after the conversion of NH$_4^+$ to N$_2$ by the addition of alkaline hypobromite (Warembourg, 1993). The δ$^{13}$C-DIC samples for enrichment assessment were analysed with IRMS (Thermo Scientific Delta V Advantage) coupled with Finnigan Gasbench II (Thermo Fisher Scientific). The preparation of the samples and the measurements of δ$^{13}$C-DIC were carried out after Torres et al. (2005).

**2.5 Data analysis**

An analysis of variance (two-way and three-way ANOVA) was used to test the significance of differences in process rates between sites, seasons or light conditions. Due to the lack of measurement of C-fixation at the northern site, a paired t-test was used to test for significant differences in these data. Depending on the context both ANOVA and paired t-tests were employed to examine variations in water parameters between the two seasons. The assumptions of normality and homogeneity of variance

were checked using Shapiro–Wilk test and Cochran's tests, respectively. In the case of heteroscedasticity, data were square log(1+x$^2$) or square root (sqrt) transformed. For significant factors, post hoc pairwise comparisons were performed using the Student-Newman-Keuls (SNK) test. The significance level was set at α = 0.05.

Riverine NO$_3^-$ loads to the lagoon were derived from continuous measurements of river discharge (obtained from the Lithuanian Hydrometeorological Service) and periodic measurements of riverine DIN concentrations. River samples were

collected at approximately monthly intervals, with supplemental samples collected during periods of high discharge (March–April). Approximately 150 measurements of DIN were obtained at the river gauging station during 2012–2021. To infer daily concentrations, we modelled seasonal, inter-annual and discharge-dependent variation in riverine DIN concentrations using a Generalized Additive Model (Bukaveckas et al., 2023). The models were used to predict daily concentrations in the river, and, in combination with daily discharge, to derive daily loading values for the lagoon. Daily riverine loads were divided by the

area of the lagoon to estimate the daily areal load. The average DIN load during the spring (March–May) and summer (June–August) of 2022 were used to provide a context for average rates of NO$_3^-$ nitrate cycling within the lagoon.

Results are given as average values and standard errors. Graphical work was performed using SigmaPlot 14.0. All data are given with mean values and standard errors based on replicates.



# 3 Results

## 3.1 Spring vs. summer environmental conditions

At both stations, the water column was well-mixed, well-oxygenated (over 90% air-saturation) and of low salinity (<0.3 SU), except in summer (1.03 PSU), when seawater entered the northern part of the lagoon (Table 1). At the northern and the central sites, temperatures were lower (14.9 °C vs. 20.5 °C on average; t-test, t=-8,7, p = 0.036) and $NO_3^-$ concentrations higher (173.5 ± 30.4 µmol L$^{-1}$ vs. 0.38 ± 0.01 µmol L$^{-1}$; two-way ANOVA after sqrt transformation, $F_{1,11}$=13716.5, p <0.001) in spring than in summer. In spring, the concentration of $NO_3^-$ was higher at the northern (riverine) site (SNK test, p < 0.05) in comparison to the south-central (confined) site (Table 1). As water temperature increased in summer, $NO_3^-$ concentrations decreased in the lagoon.

**Table 1. Measured *in situ* environmental variables. kd – extinction coefficient of the light in the water column. The values show the mean ± standard error based on three replicates except for the central site, where 6 replicates were averaged based on surface and bottom water measurements.**

| Measure | Units | Spring season | | Summer season | |
|---|---|---|---|---|---|
| | | Northern site | Central site | Northern site | Central site |
| kd | [m$^{-1}$] | 1.42 | 1.21 | 2.02 | 2.39 |
| Depth of euphotic zone | [m] | 2 | 3 | 1.5 | 1.5 |
| Temperature | [ºC] | 15.0 | 14.7 | 20.0 | 21.0 |
| Salinity | [PSU] | 0.11 | 0.24 | 1.03 | 0.23 |
| $NH_4^+$ | [µmol L$^{-1}$] | 0.24±0.04 | 0.10±0.03 | 0.71±0.12 | 0.12±0.01 |
| $NO_3^-$ | [µmol L$^{-1}$] | 241.6±1.2 | 105.4±0.4 | 0.38±0.03 | 0.37±0.02 |
| DIN | [µmol L$^{-1}$] | 243.4±1.2 | 106.4±0.4 | 1.15±0.14 | 0.55±0.02 |
| DON | [µmol L$^{-1}$] | 31.8±2.5 | 15.6±1.9 | 44.0±2.0 | 38.8±0.6 |
| PON | [µmol L$^{-1}$] | 31.2±6.1 | 39.7±1.2 | 64.7±2.7 | 54.0±1.8 |
| C:N in seston | [molar] | 6.8±0.1 | 6.6±0.1 | 6.8±0.1 | 6.4±0.2 |
| Chlorophyll a | [µg L$^{-1}$] | 26.6±0.2 | 29.6±0.9 | 46.3±0.7 | 48.8±2.8 |

In contrast, the concentration of DON showed a significant increase from spring to summer at both sites (two-way ANOVA, $F_{1,11}$=10.2, p = 0.013). DON consistently remained above 10 µmol L$^{-1}$ throughout the entire study period and constituted 98% of TDN in the summer. Similarly, the concentration of PON increased from spring (range 20.5–39.9 µmol L$^{-1}$) to summer (range 51.4–69.8 µmol L$^{-1}$) (two-way ANOVA, $F_{1,11}$=7.8, p = 0.023) with significantly higher (SNK test, p < 0.05) concentrations at the northern site (Table 1). The changes in PON concentration were closely linked to the seasonal patterns of Chl-a (Table 1), which increased from spring to the summer at both sites (two-way ANOVA, $F_{1,11}$=40.2 and $F_{1,11}$= 2079.6, p < 0.001, respectively in northern and south-central sites).

During spring, the phytoplankton community mainly consisted of diatoms, which accounted for up to 77% of the total biomass (Fig. 2). The dominant diatom species were *Diatoma tenuis* and *Stephanodiscus hantzschii*. In summer, diatoms were replaced




by cyanobacteria, which accounted for nearly 75% of the total phytoplankton biomass. The dominant cyanobacteria species were non-heterocytous *Planktothrix agardhii* followed by heterocystous *Aphanizomenon flos-aquae*. Overall $N_2$-fixing cyanobacteria (predominantly *A. flos-aquae*, *Dolichospermum affine*, and *Cuspidothrix issatschenkoi*) accounted for 14–30%

of the cyanobacteria biomass. Light attenuation increased from spring to summer following Chl-a dynamics. The depth of the photic zone decreased by half, causing the sediments in the shallower (northern) lagoon area to be shaded (aphotic) during the summer (Table 1). Sediments in the deeper central-southern area were aphotic during both spring and summer.

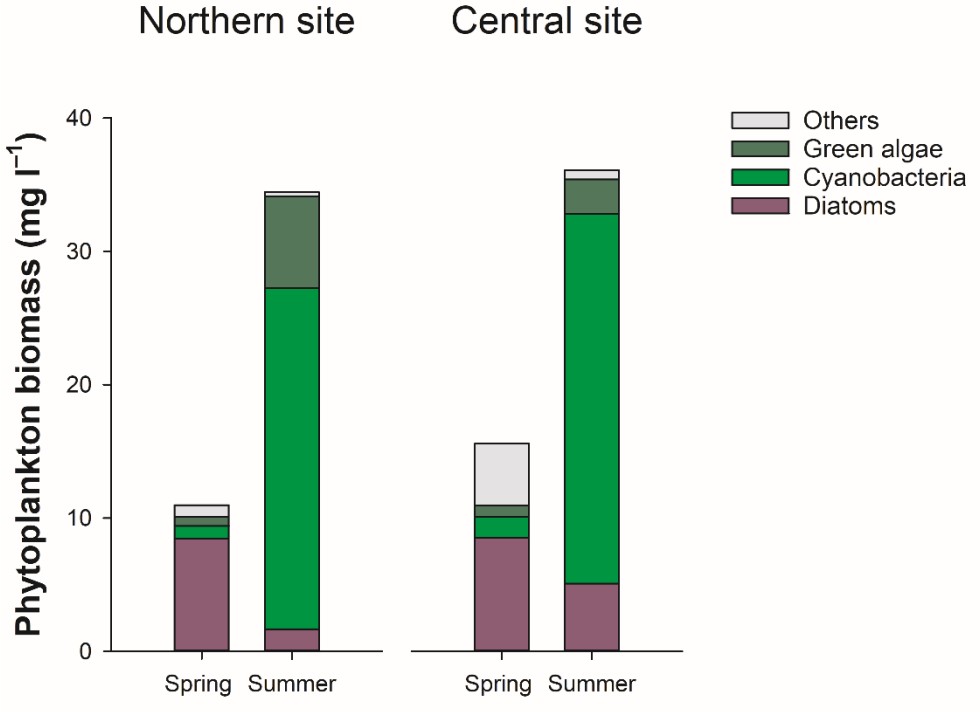

**Figure 2. Phytoplankton biomass (n=1) at the central and northern and central sites in the Curonian Lagoon during spring (May) and Summer (August) 2021.**

Discharge of the Nemunas River followed expected seasonal patterns with the highest flows occurring in spring and winter (926 $m^3$ $s^{-1}$), and lowest flows in summer and fall (300 $m^3$ $s^{-1}$, Fig. 3). DIN was the dominant fraction accounting for nearly 80% of TN in riverine inputs. $NO_3^-$ comprised most of the DIN load with concentrations from 0.05 to 364 µmol $L^{-1}$, with the

highest concentration in spring.





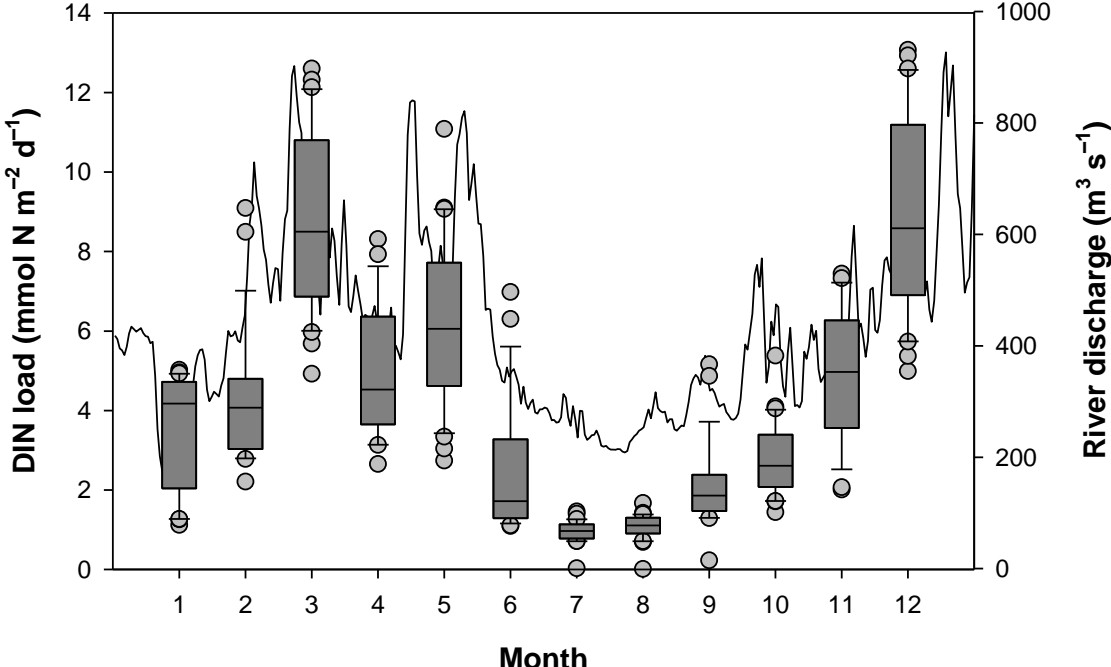

**Figure 3. Daily discharge of the Nemunas River and riverine dissolve inorganic nitrogen (DIN) load to the Curonian Lagoon during 2021. Data range (whiskers), upper and lower quartiles (edges), the median (horizontal line), and the outliers (grey circle) are represented (n=28–31).**


## 3.2 Net $NO_3^-$ uptake and C-fixation in the water column

Positive $NO_3^-$ uptake rates were observed in both light and dark bottle incubations (Fig. 4A,C). Significant differences were found between spring and summer, stations, and light vs. dark incubations (three-way ANOVA, $F_{1,23}$=7.4, p=0.015). Net $NO_3^-$ uptake increased from spring to summer with significantly higher (SNK test, < 0.05) rates at the central site (0.218 ± 0.004

µmol N $L^{-1}$ $h^{-1}$) compared to the northern site (0.120 ± 0.002 µmol N $L^{-1}$ $h^{-1}$). At both sites, net $NO_3^-$ uptake in the light (range 0.073–0.248 µmol N $L^{-1}$ $h^{-1}$) exceeded (SNK test, < 0.05) that in the dark (range 0.018–0.198 µmol N $L^{-1}$ $h^{-1}$). Water column integrated results show that during the spring net daily $NO_3^-$ uptake was low (<2 mmol N $m^{-2}$ $d^{-1}$, Fig. 4B,D). Higher areal rates were measured in summer, particularly at the south-central site (~15 mmol N $m^{-2}$ $d^{-1}$).



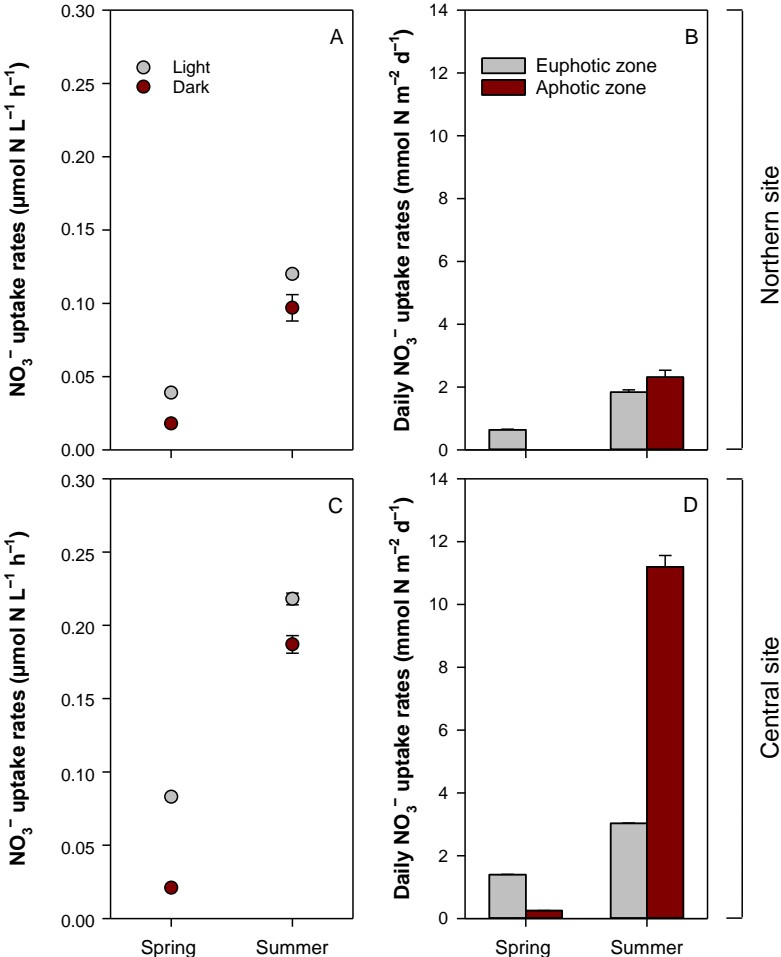


**Figure 4. Volumetric nitrate uptake rates (A,C) in the water column under photic (Light) and aphotic (Dark) conditions, and daily integrated rates (B,D) in euphotic and photic zones at two sites in the Curonian Lagoon in spring (May) and summer (August) 2021. Data shown are mean values and standard errors (n=3).**

The highest volumetric rates of C-fixation were found at the central site, which showed an increasing trend from spring to

summer (two-way ANOVA, $F_{1,11}=1300.6$, $p < 0.001$) (Fig. 5A,C). In summer, C-fixation at this site was $16.1 \pm 0.3$ µmol C L$^{-1}$ h$^{-1}$, whereas at the northern site, C-fixation was 7-fold lower ($1.9 \pm 0.4$ µmol C L$^{-1}$ h$^{-1}$; t-test, t=28.9, p < 0.001). C-fixation in the dark was low in all incubations (<0.5 µmol C L$^{-1}$ h$^{-1}$). Daily- and water column-integrated rates varied from 26.8 to 183.3 mmol C m$^{-2}$ d$^{-1}$ with the highest rates in summer at the central site (Fig. 5B,D). Rates of C-fixation during the daytime were used to estimate phytoplankton N demand. The N demand increased 4-fold from spring ($5.6 \pm 0.2$ mmol N m$^{-2}$ d$^{-1}$) to



summer ($22.3 \pm 0.4$ mmol N m$^{-2}$ d$^{-1}$) at the central site. During the summer season, the estimated phytoplankton N demand at

the northern site ($3.1 \pm 0.6$ mmol N m$^{-2}$ d$^{-1}$) was considerably lower compared to the central site (no spring data for this site).

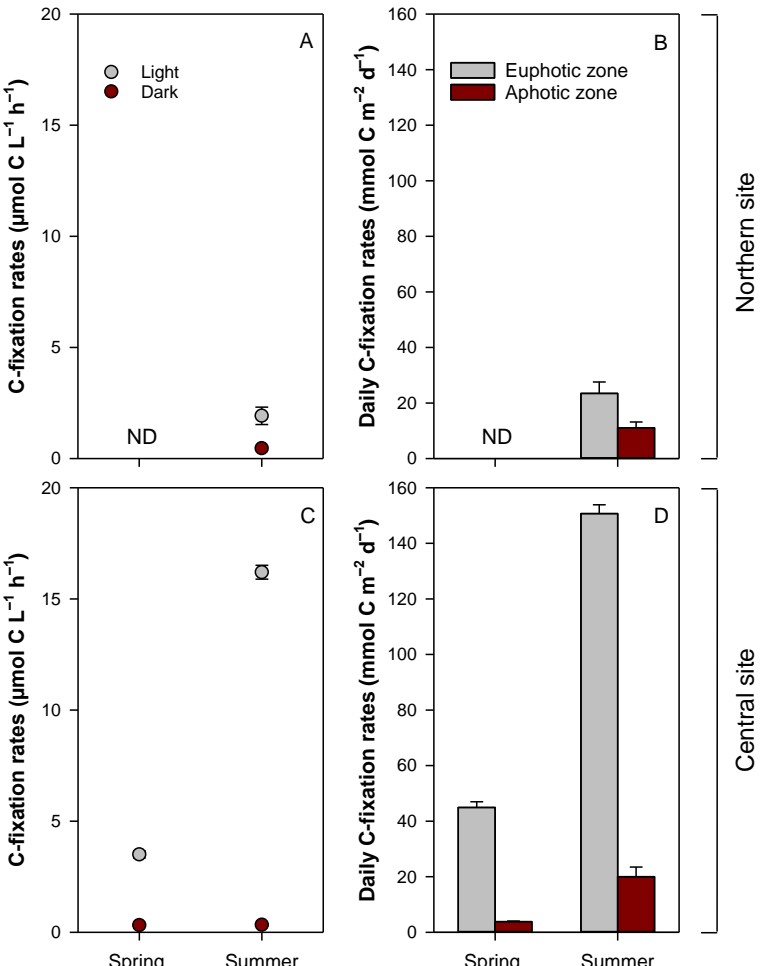

**Figure 5. Volumetric carbon fixation rates (A,C) in the water column under photic (Light) and aphotic (Dark) conditions, and daily integrated rates (B,D) in euphotic and photic zones at two sites in the Curonian Lagoon in spring (May) and summer (August) 2021.**

**Data shown are mean values and standard errors (n=3); ND – no available data.**

## 3.3 Benthic NO$_3^-$ fluxes and dissimilatory reduction processes

At both sites, rates of benthic processes were generally lower when compared to pelagic processes. Net NO$_3^-$ fluxes at the

sediment–water interface varied from –5.4 to 2.2 mmol N m$^{-2}$ d$^{-1}$ at the two studied sites. Overall, measured net NO$_3^-$ fluxes





differed only between seasons (two-way ANOVA after log transformation, $F_{1,19}$=26.1, p < 0.001), with no statistically significant differences between the two sites. In spring, sediments acted as a sink for $NO_3^-$ from the overlaying water column, with a flux of –2.0 ± 0.7 mmol N $m^{-2}$ $d^{-1}$ (averaged data across sites), however, in summer, there was an efflux of 0.01 ± 0.05 mmol N $m^{-2}$ $d^{-1}$ (averaged data across sites) (Fig. 6A,D).

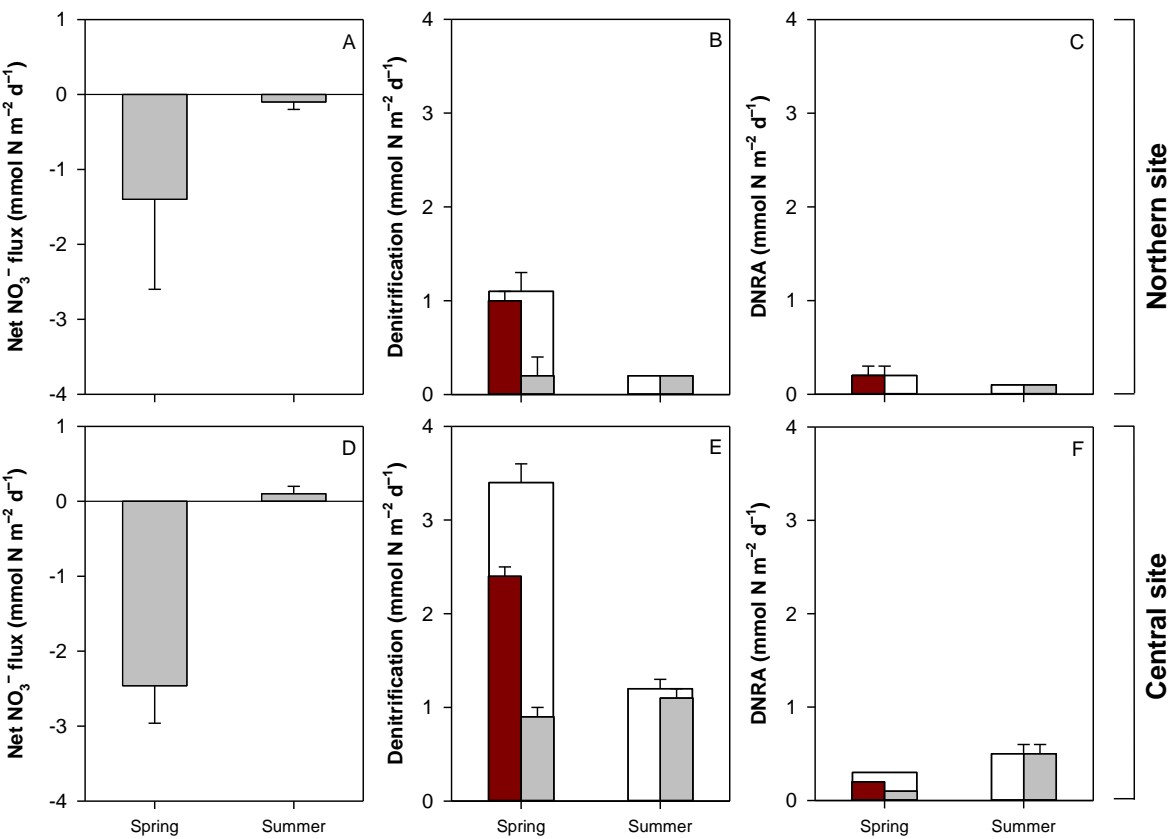


**Figure 6. Net $NO_3^-$ fluxes (A,D), total denitrification ($D_{tot}$; B,E), denitrification fueled by water column $NO_3^-$ ($D_w$) and coupled to nitrification ($D_n$), and total dissimilative nitrate reduction to ammonium ($DNRA_{tot}$; C,F), DNRA fueled by water column $NO_3^-$ ($DNRA_w$) and coupled to nitrification ($DNRA_n$) measured at the two sites in the Curonian Lagoon in spring (May) and summer (August) 2021. The positive and negative values of $NO_3^-$ fluxes indicate the release and uptake of nutrients, respectively. Data shown**
**are mean values and standard errors (n=5).**

The highest total denitrification rates ($D_{tot}$; 2.3 ± 0.4 mmol N $m^{-2}$ $d^{-1}$ averaged data) were measured in spring at both sites compared to the summer (two-way ANOVA, $F_{1,19}$=310.3, p < 0.001) (Fig. 6B,E). The decrease in $D_{tot}$ rates was attributed to the decreased levels of $NO_3^-$ in the overlaying water column, which affected $D_w$ that relies on diffused $NO_3^-$ from bottom



water. As a result, $D_w$ rates decreased from 0.7–2.7 mmol N m$^{-2}$ d$^{-1}$ to <0.1 mmol N m$^{-2}$ d$^{-1}$ from spring to summer. While the

rates of coupled nitrification–denitrification ($D_n$) remained similar during both seasons (range 0.1–1.3 mmol N m$^{-2}$ d$^{-1}$), the

overall contribution to total $N_2$ production increased from 14% in spring to 98% in summer. Higher rates were found in the

muddy sediments at the south-central site.

Rates of total DNRA varied significantly between sites and seasons (two-way ANOVA, $F_{1,19}$=22.6, p < 0.001) (Fig. 6C,F).

Similarly to denitrification, DNRA in sediments was primarily (~80%) fueled by $NO_3^-$ from overlaying the water in spring

($DNRA_w$), while in summer it was coupled to nitrification ($DNRA_n$). $DNRA_w$ varied between < 0.01 and 0.4 mmol N m$^{-2}$ d$^{-1}$

without significant differences between the two sites (two-way ANOVA, $F_{1,19}$=32.7, p < 0.001). $DNRA_n$ rates ranged between

0 and 0.6 mmol N m$^{-2}$ d$^{-1}$ with higher rates (SNK test, p < 0.05) in muddy sediments during the summer (0.5 ± 0.1 mmol N

m$^{-2}$ d$^{-1}$). At both sites, ~20% of total benthic $NO_3^-$ reduction was attributed to $DNRA_{tot}$.

In contrast to the deeper central area, sediments in the shallower northern site were illuminated in spring, which had a

significant effect on dissimilatory $NO_3^-$ processes. During light exposure, the total denitrification rates on an hourly basis

decreased by 32% compared to the rates in the dark (paired t-test, t=2.9, p = 0.022). This decrease was attributed to the

suppression of $D_w$ and $D_n$ (Table 2). In contrast, DNRA was significantly (paired t-test, t=-3.1, p = 0.02) stimulated by

illumination, with ~90% higher rates in light incubations compared to those in the dark.

**Table 2. Hourly rates (µmol N m$^{-2}$ h$^{-1}$) of benthic dissimilatory nitrate ($NO_3^-$) reduction through denitrification and dissimilative nitrate reduction to ammonium (DNRA) under different light conditions in spring at northern (sandy) and central (muddy) sites in the Curonian Lagoon. $D_{tot}$ – total denitrification, $D_w$ – denitrification fueled by $NO_3^-$ from the overlaying water column, $D_n$ – coupled nitrification-denitrification, $DNRA_w$ – DNRA fueled by $NO_3^-$ from the overlaying water column, $DNRA_n$ – coupled nitrification-DNRA. Data shown are mean values and standard errors (n=5).**


| $NO_3^-$ reduction process | | Central site | | Northern site | | |
|---|---|---|---|---|---|---|
| | | Spring | Summer | Spring | | Summer |
| | | Dark | Dark | Light | Dark | Dark |
| Denitrification | $D_{tot}$ | 141.0±5.4 | 48.4±2.6 | 24.8±4.1 | 77.0±15.3 | 6.7±0.9 |
| | $D_w$ | 101.9±2.4 | 1.3±0.6 | 22.5±6.4 | 65.0±1.0 | 0.1±0.0 |
| | $D_n$ | 39.1±1.7 | 47.2±2.8 | 2.4±3.0 | 12.0±2.9 | 6.6±0.9 |
| | | | | | | |
| DNRA | $DNRA_{tot}$ | 12.7±1.0 | 21.9±1.6 | 14.9±3.8 | 1.8±0.1 | 2.3±0.3 |
| | $DNRA_w$ | 9.2±0.8 | 0.6±0.3 | 13.2±3.9 | 1.5±0.2 | <0.05 |
| | $DNRA_n$ | 3.5±0.2 | 21.3±1.5 | 1.7±1.4 | 0.3±0.0 | 2.3±0.3 |





## 4 Discussion

### 4.1 Seasonal factors affecting pelagic $NO_3^-$ uptake

In northern latitudes, during the spring season, the presence of excess N and lower temperatures generally favor the rapid
growth of diatoms, which have a high affinity for $NO_3^-$ (Lomas and Glibert, 2000; Berg et al., 2003; Olofsson et al., 2018).
The transition from diatom-dominated to cyanobacteria-dominated communities in summer was associated with a marked
increase in pelagic $NO_3^-$ uptake because most of the cyanobacteria encountered were non-diazatrophic species. This increase
can be attributed in part to the substantially higher phytoplankton biomass observed during summer (as indicated by Chl-a).
Even when $NO_3^-$ uptake rates are normalized relative to biomass (using PON as a proxy), we found that biomass-specific
uptake rates were 65–86% higher in summer. The increase in biomass-normalized rates of $NO_3^-$ uptake occurred despite low
DIN concentrations in the lagoon during summer. This suggests that cyanobacteria may possess a nutritional strategy to
efficiently utilize $NO_3^-$ despite a preference for $NH_4^+$ (Chaffin and Bridgeman, 2013). A comparison of $NO_3^-$ uptake rates with
estimated phytoplankton N demand (from C fixation measurements) shows that the contribution of $NO_3^-$ to the estimated
phytoplankton N demand was relatively small in both spring and summer (~22% on average across sites). This suggests that
$NH_4^+$ or DON assimilation or dinitrogen ($N_2$) fixation satisfied most nutritional needs for the growth of phytoplankton.
However, this estimation is rather speculative as it only considers the euphotic layer and estimates total autotrophic demand
while in summer most processes occur under the dark conditions.

While pelagic $NO_3^-$ uptake in spring is replenished by riverine inputs, uptake in summer would rapidly (~3 hours) deplete the
small standing pool in the water column without continuous replete via organic matter mineralization and nitrification. These
cycling pathways in the water column are likely supported by the abundance of cyanobacteria. Unlike fast-sinking diatoms,
buoyant cyanobacteria remain suspended in the water column. This allows them to eventually contribute to N recycling, which
involves processes such as mineralization and nitrification (Peng et al., 2017; Hampel et al., 2019; Chen et al., 2020). We
estimated that nitrification in the water column may contribute approximately 11 mmol $NO_3^-$ $m^{-2}$ $d^{-1}$ taking into account the
average water column depth of 3.5 m. Based on the fact that the upscaled net $NO_3^-$ uptake in summer was around 9.7 mmol N
$m^{-2}$ $d^{-1}$ (Fig. 7) and low concentration prevailed, fast $NO_3^-$ turnover processes likely occurred. We estimated that roughly one-
third of the regenerated $NH_4^+$ (~30 mmol N $m^{-2}$ $d^{-1}$ as reported by Zilius et al., 2018) has the potential to undergo oxidation
during nitrification. A higher concentration of DON compared to DIN during summer suggests that organic N could also
potentially play an important maintaining N cycling in the water column through phytoplankton release, reminilazation to
$NH_4^+$ and oxidation or uptake (Wanicke et al., 2009; Korth et al., 2011; Wood and Bukaveckas, 2014; Zilius et al., 2018;
Klawonn et al., 2019).





## 4.2 Factors influencing benthic processes

The current study uncovers that benthic processes exhibited seasonal patterns that were opposite to those observed in the pelagic zone. The maximum rates in benthic $NO_3^-$ dissimilative processes were recorded in spring. This differs from other coastal areas around the Baltic Sea where dissimilative processes reach a peak in summer but are suspected to be fuelled by

organic matter from earlier in the year (Bonaglia et al., 2014; Bartl et al., 2019; Hellemann et al., 2020). Different patterns across coastal settings could be explained by variable $NO_3^-$ and labile organic matter availability (Hietanen and Kuparinen, 2008; Zilius et al., 2018; Bartl et al., 2019). The higher pelagic $NO_3^-$ concentrations and the accumulation of settling diatoms, are likely the primary drivers of high denitrification rates in spring (2.3 mmol N m$^{-2}$ d$^{-1}$), which exceed other coastal areas in the Baltic Sea (0.2 mmol N m$^{-2}$ d$^{-1}$ Gulf of Finland, Hietanen and Kuparinen, 2008; 0.2 mmol N m$^{-2}$ d$^{-1}$ in Himmerfjärden

estuary, Bonaglia et al., 2014; 0.2 mmol N m$^{-2}$ d$^{-1}$ in Archipelago, Hellemann et al., 2020; 1.1 mmol N m$^{-2}$ d$^{-1}$ in Öre Estuary, Zilius et al,. 2021). DNRA which is rarely measured in lagoons and estuaries around the Baltic was in a similar range in the Curonian Lagoon (0.2 mmol N m$^{-2}$ d$^{-1}$) and in the Öre Estuary (0.3 mmol N m$^{-2}$ d$^{-1}$, Zilius et al. 2021), in the Himmerfjärden estuary (0.1 mmol N m$^{-2}$ d$^{-1}$, Bonaglia et al., 2014) and in the Gulf of Finland's archipelago (0.1 mmol N m$^{-2}$ d$^{-1}$, Hellemann et al., 2020). This suggests that denitrification is the main driver of DIN concentrations and therefore the most important

process in the benthic–pelagic coupling.

The findings reveal that during summer months when there is a decrease in pelagic $NO_3^-$ concentrations, the main processes responsible for the $NO_3^-$ production are mineralization and nitrification in sediments. However, the ammonification rates in the sediments (up to 5 mmol N m$^{-2}$ d$^{-1}$; Zilius et al., 2018) are six times lower compared to the turnover in the water column. In the south-central area of the lagoon, where organic-rich deposits accumulate, higher rates of mineralization and nitrification,

result in higher rates of dissimilatory $NO_3^-$ processes during the summer. Such internal $NO_3^-$ turnover via linked microbial processes is the main mechanism in other coastal settings where $NO_3^-$ is typically below <10 µmol L$^{-1}$ throughout the year (Hellemann et al., 2020; Zilius et al., 2021). The prevalence of denitrification over DNRA in these sediments is likely influenced by lower salinity levels (Giblin et al., 2010). Nevertheless, when brackish water intrusion leads to higher salinity, reductive processes could cause the accumulation of sulfide or reduced metal forms, thereby facilitating DNRA (Kessler et al.,

2018; Murphy et al., 2020).

Microphytobenthos or settled diatoms can also play a significant role in the assimilation of $NO_3^-$ (Fig. 7). This process mainly occurs during the spring in the shallower (northern) half of the lagoon where the benthic assimilative pathway accounts for approximately 40% of denitrification ($D_w$). Depending on the light conditions, N uptake by microalgae in shallow sandy sediments can frequently exceed the amount loss via denitrification, and photosynthetic microorganisms appear to inhibit

denitrification (Sundbäck et al., 2006; Bartoli et al., 2021). Moreover, we cannot exclude the possibility that a portion of assimilated $NO_3^-$ by diatoms is later respired through dissimilatory pathways, as suggested study by Merz et al. (2021). In



general, the transformation of $NO_3^-$ in sediments becomes insignificant during the summer months compared to the water column.

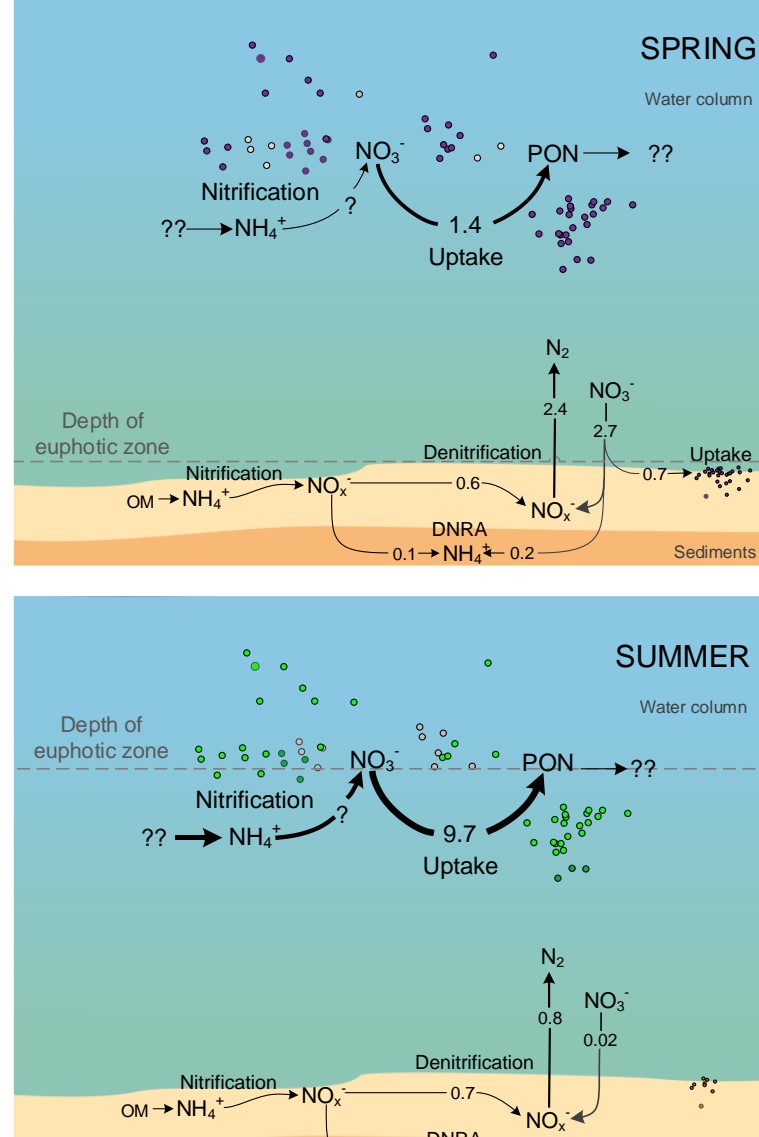

**Figure 7. Flowchart of N-cycling in the Curonian Lagoon water column and sediments in spring (during diatom bloom and N excess)
and summer (during cyanobacteria bloom and N scarcity). Nitrogen transformations were upscaled to the lagoon combining
measured fluxes and $NO_3^-$ reduction processes at dominant macro areas. Note that rates reported in the figure are expressed as
mmol N $m^{-2}$ $d^{-1}$. The circle symbols indicate the relative abundance of dominant pelagic groups: diatoms (purple), cyanobacteria
(light green), green algae (dark green), and others, including heterotrophs (grey).**




### 4.3 Pelagic community shadows water column and regulates process rates

Throughout the year, the turbidity in the water column of the lagoon changes, which impacts the depth of the euphotic zone (Zilius et al., 2014). Light scattering is primarily caused by buoyant cyanobacteria that reduce the euphotic zone to ~1 m in summer. As a result, self-shaded cyanobacteria and other microorganisms experience periods of darkness. However, through the combined effects of gas vesicles and wind-driven vertical mixing, cyanobacteria remain in continuous motion, temporarily accessing the narrow euphotic zone. The results show that the rates of $NO_3^-$ uptake in light were consistently higher than those
in darkness (by 16–62%), indicating more effective uptake during the daylight hours within the euphotic zone. Previous studies have shown that some phytoplankton groups (i.e. diatoms) can also assimilate $NO_3^-$ in the dark (Lomas and Glibert, 1999). Based on our previous metagenome analyses in the lagoon, the presence in the summer of the marker genes responsible for assimilative $NO_3^-$ reduction was mainly attributed to Cyanobacteria, Firmicutes and Euryarcheota (Broman et al., 2021). This implies that the cyanobacterial community is capable of assimilating $NO_3^-$ throughout the entire water column, even below
the euphotic zone, and thus competing with heterotrophs (Chaffin and Bridgeman, 2014; Hampel et al., 2019). In the summer, the dominant *Planktotrix* genus is adapted to lower light intensities than other cyanobacteria, making it more competitive (Post et al., 1985).

A striking result from our study is the contrasting impact of light conditions on benthic processes. Sedimentary denitrification, which was fuelled by $NO_3^-$ from the water above, decreased in the presence of light. This decrease can be
attributed to the fact that microphytobenthos are photosynthetic active. As a result, oxygen is produced and diffuses into the sediment, creating an oxic zone and, thus, extending the diffusion pathway for $NO_3^-$ to the denitrification zone (Risgaard-Petersen et al., 2005). Additionally, settled diatoms and other microphytobenthos assimilate $NO_3^-$ (Stief et al., 2022) leading to a competition between algae and heterotrophic denitrifiers (Risgaard-Petersen, 2003; Sundbäck et al., 2006). In contrast, light has been found to enhance DNRA, which is not commonly observed in intact core incubations (e.g. Dunn et al., 2012).
The increased total DNRA rates are attributed to an increase in the uptake of $^{15}NO_3^-$ ($DNRA_w$) from the water column. There is evidence to suggest that eukaryotic phototrophs, such as diatoms, can respire assimilated $NO_3^-$ in deeper sediment layers without oxygen and $NO_3^-$ (Merz et al., 2021). Currently, this N cycling pathway is not well understood, but it could be that settled diatoms together with prokaryotic microorganisms may drive DNRA (Stief et al., 2022). In spring, the photic zone expands up to a depth of 3 m, indicating that approximately 80% of the lagoon's sediment area is exposed to light. This light
exposure may help synchronize $NO_3^-$ reduction and $NH_4^+$ production in these microorganisms, aligning with those times of the day when microphytobenthos or settled diatom cells are photosynthesizing and can utilize $NH_4^+$ or gain energy through $NO_3^-$ respiration.

### 4.4 Importance of $NO_3^-$ cycling in the context of the lagoon's N budget

Simultaneous measurements of pelagic and benthic $NO_3^-$ processes allowed us to depict cycling and its contribution to $NO_3^-$
retention during two contrasting seasons (Fig. 7). During the spring season, the average pelagic assimilation of $NO_3^-$ (1.2 mmol



N m$^{-2}$ d$^{-1}$), derived from the upscaled site-specific measurements accounting for the northern (45%) and south-central areas (55%) (Zilius et al., 2018), was estimated to equal to 18% of average daily N load (6.6 mmol N m$^{-2}$ d$^{-1}$ in March–May) delivered to the lagoon. In summer, pelagic demand for NO$_3^-$ increased and exceeded by 85% riverine inputs to the lagoon (1.5 mmol N m$^{-2}$ d$^{-1}$). Higher NO$_3^-$ uptake in the water column in the south-central area was due to more biomass, which may

reflect longer water renewal time in comparison to the northern site (Zilius et al., 2014; Vaičiūtė et al., 2021). In spring, water renewal time in the lagoon is approximately 60 days (Umgiesser et al., 2016), which enables effective NO$_3^-$ transformations through assimilative and/or dissimilative processes. Consequently, prolonged retention of river water within the lagoon will lead to a higher retention of NO$_3^-$ (Dettmann, 2001). In addition, NO$_3^-$ retention is influenced by seasonal changes in dominant phytoplankton communities as diatom assimilated N settles in surface sediment whereas cyanobacteria keeps PON suspended

in the water column, leading to its recycling or export to the Baltic Sea (Zilius et al., 2018). Overall, NO$_3^-$ assimilation, especially in summer when N limation exists, represents an important N cycling pathway in the water column after NH$_4^+$ assimilation (Broman et al., 2021). This demand for NO$_3^-$ far exceeds N inputs via biological dinitrogen fixation (2.1 mmol N m$^{-2}$ d$^{-1}$; Zilius et al., 2021) as well as NO$_3^-$ release from sediments and river inputs. However, it remains questionable whether nitrification in the water column can sustain such NO$_3^-$ demand. Therefore, future studies should also include measurements

of NO$_3^-$ production via nitrification and other cycling pathways (Fig. 7).

In spring, upscaled net daily NO$_3^-$ uptake by sediments (2.7 mmol N m$^{-2}$ d$^{-1}$) was on average equivalent to 41% of the N load (Fig. 7). Our results further indicate that denitrification (D$_w$) alone accounted for 27% of N load, which was permanently removed from the ecosystem. This contribution is significant when compared to the other coastal settings around the Baltic Sea (16%, Asmala et al. 2017). Despite decreasing riverine N loads in summer, the importance of D$_w$ also decreased and was

equivalent only to 1% of the load. We attribute low denitrification rates in summer to depletion of NO$_3^-$ concentrations in the water column, primarily resulting from phytoplankton assimilation. In addition, we found that 30% of NO$_3^-$ retained by sediments in spring underwent other transformations, including assimilation by microalgae and reduction via DNRA.

The previous mass balance estimations by comparing inputs and outputs to the system (60% of load retention) conducted by Vybernaite-Lubiene et al. (2017) support the estimations presented herein. The total retained fraction of riverine inputs (59%)

through the multiple NO$_3^-$ cycling pathways in the Curonian Lagoon exceeds these in other coastal settings (e.g. 12–29% in the Oder Lagoon, Grelowski et al. 2000; 18% in the Vistula Lagoon, Witek et al. 2003; 19% Archipelago Sea, Silvennoinen et al. 2007) or is similar to large embayments, such as the Stockholm Archipelago (65%, Almroth-Rosell et al. 2016). In comparison, the estuarine systems in southern Europe have relatively shorter residence times such that NO$_3^-$ inputs are rapidly flushed to adjacent coastal areas (Middelburg and Nieuwenhuize 2000a).

## 5 Conclusions


In this study, we provide insights into how seasonal changes in riverine inputs, phytoplankton community composition and microbial rates influence the pelagic and benthic components of N cycling in a shallow coastal lagoon. In spring, benthic processes were important as diatom blooms and elevated riverine inputs favored high rates of denitrification and net flux of N from the water column to the sediments. In summer, cyanobacteria blooms caused high rates of pelagic assimilation, which

coupled with low riverine inputs resulted in the depletion of DIN and minimal N fluxes across the sediment–water interface. Denitrification dominated in spring while DNRA remained low in both seasons pointing to the important control of OM from settling phytoplankton. Our findings are consistent with the paradigm that eutrophication favors a shift from benthic to pelagic-dominated processes. Furthermore, these findings suggest that the greater prevalence of cyanobacteria while likely enhancing the efficiency of DIN retention within the lagoon, reduces the efficiency of total N retention because of greater export of DON

and PON, relative to periods dominated by diatom communities.

**Authors contribution**

MZ, IK and SB conceived the ideas and designed methodology. MZ, IVL, EL and TB led the field survey and experimental

activities. IV-L, EL, RB and TB assisted with analysis and data collection and analysis. MZ and SB secured funding for the investigation. MV provided use of specialized facilities. MZ and PAB wrote the first draft of the paper, and all co-authors contributed to writing review and editing.

**Conflict of interest**

The authors declare that they have no conflict of interest.

**Acknowledgment**

We are in debt for the Coast Guard District of the State Border Guard Service for logistic support. We thank kindly thank Doanata Overlinge for phytoplankton microscopic analysis and Jovita Mėžinė for map design.

**Financial support**

the "Unravelling hidden players and pathways of nitrogen cycling in the three largest European lagoons (CycloN)" (Agreement

No. S-MIP-22-47) grant under agreement with the Research Council of Lithuania (LMTLT). IV-L and TP were also supported by the Mikronitro project, grant No. 28T-2021-36/SUT-21P-11. MV was supported by the BluEs project, grant No. 03F0864A.



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
