# Peer review of "Riverine inputs and phytoplankton community composition control nitrate cycling in a coastal lagoon"

_EGUsphere, 2023_

## Referee Comment (RC1)

[referee-annotated manuscript omitted]

---

## Author Comment (AC1)

**Response to Anonymous Referee #1**

**General comments**

The Ms on "Riverine inputs and phytoplankton community composition control nitrate cycling in a coastal lagoon" by M. Zilius is a well written manuscript. It is presented in a concise manner, solid methods & results, and appropriate discussion and conclusion. I have some minor remarks in the text (attached pdf file) and two questions for the discussion.

The study focusses on the spring and summer situation. But what is known or is expected to happen during the rest of the year?

What are exchange processes to the Baltic Sea? Where is the connecting channel in Figure 1? and in relation to the river discharge how much water is leaving or entering the lagoon, and what would this mean for N-export ...?

**Answer:** Thank you for the positive comments.

This study builds on our earlier work documenting nutrient mass balances in the Curonian Lagoon (Vybernaite-Lubiene et al., 2017, 2022; Zilius et al., 2018) by quantifying specific pathways of N transformation. We acknowledge that some additional background information about our prior work needs to be included for readers to understand why we chose specific seasons and sampling locations for the present work. We are also in agreement with the reviewer that more background information should be provided regarding N transport into the lagoon and exchange processes with the Baltic Sea. To address these issues we have: (1) added a new paragraph (preceding the last section of the Introduction) that describes the results of the previous mass balance work, including seasonal patterns, and (2) added information to the Site Description describing the hydrology of the lagoon and dominant exchange processes. Text added to Introduction:

*"A recent study examining input-output fluxes of the Curonian Lagoon (SE Baltic Sea), showed that $NO_3^-$ concentrations in outflowing water were significantly lower relative to riverine inputs (Vybernaite-Lubiene et al., 2017). Attenuation of $NO_3^-$ in estuarine systems is typically attributed to phytoplankton or bacteria assimilation. Following the conversion of dissolved inorganic nitrogen (DIN) to particulate organic nitrogen (PON), there are two potential pathways for assimilated N: (1) enhanced export of PON (resulting in minimal net N retention), or (2) sedimentation of PON (resulting in net N retention via storage or denitrification). Our prior work on the Curonian Lagoon suggests that sedimentation is the dominant fate of assimilated N in the spring, whereas remineralization and export of PON was the dominant pathway in summer. We hypothesized that seasonal differences in N pathways were dependent on the composition of the phytoplankton community, which can lead to deposition in sediments (in the presence of heavier diatoms) or PON export and remineralization (in the presence of buoyant cyanobacteria). The aim of our further work was to quantify these pathways during diatom vs. cyanobacteria dominated conditions.*"

Text added to Study site description:

*"The Curonian Lagoon is a large (1584 $km^2$), shallow (mean depth 3.8 m) waterbody located along the southeast coast of the Baltic Sea (Fig. 1). The lagoon is mainly freshwater (mean salinity = 0.2 PSU) due to large riverine inputs and limited exchange with the Baltic Sea through a narrow channel (Zemlys et al., 2013). The lagoon is vertically well-mixed owing to the shallow depth and weak salinity gradients (Zilius et al., 2014, 2020). The Nemunas River is the principal tributary (16 $km^3$ $yr^{-1}$, Vybernaite-Lubiene et al., 2018) accounting for 96% of total water inputs and the main source of nutrients (Jakimavičius and Kriaučiūnienė, 2013; Vybernaite-Lubiene et al., 2022). On average, riverine inputs (46.2 × $10^6$ $m^3$ $d^{-1}$) are ~10% lower than lagoon outflow (50.6 × $10^6$ $m^3$ $d^{-1}$) due to prior seawater inflow to the lagoon (Vybernaite-Lubiene et al., 2022)."*

The lagoon discharges to the Baltic Sea through a narrow strait (see below corrected Figure 1) and it occasionally receives inputs from the Baltic during periods of wind-driven tidal forcing (Zemlys et al. 2013). These events are typically of short duration and result in small increases in salinity (typically by 1–2, maximum = 7) in the northern portion of the lagoon.

Line 95 There are no red / blue circles visible…. Could you also indicate the Connection to the Baltic Sea?

**Answer:** The reviewer is right; we have correct figure captions. Following the reviewer's comment, we added an additional panel indicating lagoon outflow and the exchange site with the sea.

[Figure]

*Figure 1 Captions. Satellite image by OLI/Landsat-8 (18/09/2014) showing summer blooms in the Curonian Lagoon with the sampling sites (stars) representing the northern and south-central regions and monitoring site at the Nemunas River (black circle). The black line indicates a state border between two countries. The outflow is at the northern end of the lagoon near Klaipeda.*

Line 140 how was assured that the $O_2$ level remained within these borders? measured or estimated?

**Answer**: We have long-term experience studying benthic metabolism in this system (Zilius et al. 2012, 2014 2015; Politi et al. 2021, 2022; Bartoli et al. 2023). In addition, two randomly selected cores were equipped with optode sensor spots to monitor $O_2$ (FireSting-O2, PyroScience GmbH)

continuously. This allowed oxygen to be kept within the targeted range. We have added these details to the main text of the manuscript.

Line 231 – 234 (Figure 3). Is the load given per month?

**Answer**: Yes, we present here a monthly load of dissolved inorganic nitrogen, and following the review comment, we have corrected the figure caption:

*Figure 3. Daily discharge of the Nemunas River and monthly riverine loads of dissolved inorganic nitrogen (DIN) to the Curonian Lagoon during 2021. Data range (whiskers), upper and lower quartiles (edges), the median (horizontal line), and the outliers (grey circle) are represented (n=28–31).*

Line 271: what does the "white" column stand for?

**Answer**: The white column represents total denitrification, consisting of denitrification fueled by water column $NO_3^-$ ($D_w$) coupled with nitrification ($D_n$). The same is valid for DNRA. This is a quite common way to present denitrification rates. We added a legend, which was missing, to make the figure more explicit (see figure below).

[Figure]

*Figure 6. Net daily $NO_3^-$ fluxes (A,D), total denitrification ($D_{tot}$; B,E), denitrification fueled by water column $NO_3^-$ ($D_w$) and coupled to nitrification ($D_n$), and total dissimilative nitrate reduction to ammonium ($DNRA_{tot}$; C,F), DNRA fueled by water column $NO_3^-$ ($DNRA_w$) and coupled to nitrification ($DNRA_n$) measured at the two sites in the Curonian Lagoon in spring (May) and summer (August) 2021. The positive and negative values of $NO_3^-$ fluxes indicate the release and uptake of nutrients, respectively. Data shown are mean values and standard errors (n=5).*

---

## Author Comment (AC2)

**Response to Anonymous Referee #2**

**General comments (part 1)**

I can not recommend the publication of this work. The main interest of this manuscript is the simultaneous measurement of $NO_3^-$ uptake in the water column, due in this system to assimilation by phytoplankton and heterotrophic bacteria, and benthic $NO_3^-$ consumption, mainly due to dissimilatory processes, where $NO_3^-$ is used as a final $e^-$ acceptor in denitrification and DNRA. However, the study consists in just two sampling sites in only the north section of a very large coastal lagoon, which has been sampled just one time in spring and in summer. Therefore, both the spatial and temporal replicability is very low or inexistent (temporal). This is in my opinion a very serious drawback for the publication of this study. Otherwise, the experimental work seems well done in general, although there are some specific issues that are unclear and that should be further explained (see my specific comments below). Another big issue is that in my opinion the discussion section is very difficult to follow in general and very frequently based in statements about processes which has not been measured here. I strongly suggest to the authors avoiding what can be put in their own words in line 316 "this estimation is rather speculative as it only considers ...". For instance, all explanations of your results based diatom and cyanobacteria settling are just speculations. You could have measured yourself this rates, if you consider this important in your system. It is part of your hypothesis!

**Answer**: We appreciate the reviewer's viewpoint that the spatial-temporal replication of our study is very limited, but we hope that the reviewer will appreciate the broad range of processes that were simultaneously measured to comprehensively quantify N transformations in the lagoon. The high level of effort required for the latter precludes large scale replication of the former. It is for this reason that prior studies have picked a few of these processes to measure at a larger range of sites, or more frequently. To our knowledge, there have been no prior studies that have simultaneously measured assimilative $NO_3^-$ uptake and benthic dissimilatory processes in both pelagic and benthic habitats. As such, this study is the first to provide a comprehensive and mechanistic understanding of $NO_3^-$ cycling in an estuarine system. As the measurement of these pathways is central to understanding how riverine $NO_3^-$ loads are attenuated in estuarine systems, we feel that these findings are worthy of publication.

Our two study sites are representative of the two macro-areas in the lagoon that differ: i) in water residence time, ii) the effect of riverine inputs, and iii) sediment deposits (Ferarrin et al. 2008; Umgiesser et al. 2016; Zilius et al. 2018; Politi et al. 2023). Our two sites cover most of the lagoon sedimentary environments. Furthermore, intensive research in both macro-areas has been carried out in the last decade, delivering a solid biogeochemical basis for understanding the biogeochemical processes in the lagoon and providing the background for the manuscript. We have more explicitly explained our choice for these two sites (see text in Material and Methods):

"*The present study was carried out at two sites (Fig. 1) that are representative of the main macroareas of the lagoon (northern and central-southern), which differ: i) in water residence time, ii) the effect of riverine inputs, and iii) sediment deposits (Umgiesser et al., 2016; Zilius et al., 2014, 2018; Politi et al., 2023). The northern area is characterized by shallower depths (1.5–2 m), direct river inflow, shorter water residence time (seasonal range = 50–100 days), and sandy sediments with low organic matter content ($C_{org}$ < 0.5%) (Umgiesser et al., 2016; Zilius et al., 2018; Politi et al., 2023). The central-southern area of the lagoon is deeper (mean = 3.5 m), less affected by riverine inflow, and therefore, has a longer water residence time (seasonal range = 100–250 days), and organic-rich deposits (predominantly silty sediments, $C_{org}$ = 10–14%). The two sites cover most of the lagoon sedimentary environments (Politi et al., 2023). Furthermore, intensive research at both sites was carried out in the last decade (Bartoli et al., 2023; Zilius et al., 2014, 2018; 2020; 2021; Politi et al., 2023), delivering a solid biogeochemical basis for understanding the biogeochemical processes in the lagoon (Vybernaite-Lubiene et al., 2017; Magri et al., 2024).*"

- Ferrarin, C., A. Razinkovas, S. Gulbinskas, G. Umgiesser, L. Bliūdžiūtė, 2008. Hydraulic regime-based zonation scheme of the Curonian Lagoon. Hydrobiologia 611: 133–146. doi:10.1007/s10750-008-9454-5
- Politi, T., M. Zilius, P. Forni, A. Zaiko, D. Daunys, M. Bartoli, 2022. Biogeochemical buffers in a eutrophic coastal lagoon along an oxic-hypoxic transition. Estuarine, coastal and shelf science 279: 108132. doi:10.1016/j.ecss.2022.108132
- Umgiesser, G., P. Zemlys, A. Erturk, A. Razinkova-Baziukas, J. Mežinė, C. Ferrarin, 2016. Seasonal renewal time variability in the Curonian Lagoon caused by atmospheric and hydrographical forcing. Ocean Sciences 12: 391–402. doi:10.5194/os-12-391-2016
- Zilius, M., I. Vybernaite-Lubiene, D. Vaiciute, J. Petkuviene, P. Zemlys, I. Liskow, M. Voss, M. Bartoli, and P.A. Bukaveckas, 2018. The influence of cyanobacteria blooms on the attenuation of nitrogen throughputs in a Baltic coastal lagoon. Biogeochemistry, 141(2), 143–165, doi:10.1007/s10533-018-0508-0, 2018.
- Zilius, M., I. Vybernaite-Lubiene, D. Vaiciute, D. Overlingė, E. Grinienė, A. Zaiko, S. Bonaglia, I. Liskow, M. Voss, A. Andersson, S. Brugel, T. Politi, P.A. Bukaveckas, 2021. Spatiotemporal patterns of N2 fixation in coastal waters derived from rate measurements and remote sensing. Biogeosciences 18: 1857–1871. doi:10.5194/bg-18-1857-2021

With the data collected here, we were able to upscale processes on the lagoon scale to understand how the nutrient processes and coastal filter work. Finally, our most recent study on pelagic $N_2$ fixation confirmed that even upscaling rates from two sites in the lagoon generate meaningful results in comparison to other independent data, in this case, satellite images (Zilius et al. 2021).

Both time points represent different riverine N loadings to the lagoon and phytoplankton communities, typical for temperate lagoons in the Northern Hemisphere. Our intent was not to characterize seasonal variability in N processes, but rather, to investigate changes in N pathways during diatom- vs. cyanobacteria- dominated periods. Typically, these communities dominate during spring and summer, respectively, and that is why we chose these periods for measurement of N processes. In response to the suggestions from both reviewers, we have added a new paragraph (preceding the last section of the Introduction) that describes the results of the previous mass balance work, explains how N fluxes may be related to changes in phytoplankton community composition, and how the present study aims to resolve missing gaps:

*"A recent study examining input-output fluxes of the Curonian Lagoon (SE Baltic Sea), showed that $NO_3^-$ concentrations in outflowing water were significantly lower relative to riverine inputs (Vybernaite-Lubiene et al., 2017). Attenuation of $NO_3^-$ in estuarine systems is typically attributed to phytoplankton or bacteria assimilation. Following the conversion of dissolved inorganic nitrogen (DIN) to particulate organic nitrogen (PON), there are two potential pathways for assimilated N: (1) enhanced export of PON (resulting in minimal net N retention), or (2) sedimentation of PON (resulting in net N retention via storage or denitrification). Our prior work on the Curonian Lagoon suggests that sedimentation is the dominant fate of assimilated N in the spring, whereas remineralization and export of PON was the dominant pathway in summer. We hypothesized that seasonal differences in N pathways were dependent on the composition of the phytoplankton community, which can lead to deposition in sediments (in the presence of heavier diatoms) or PON export and remineralization (in the presence of buoyant cyanobacteria). The aim of our further work was to quantify these pathways during diatom vs. cyanobacteria dominated conditions."*

**General comments (part 2)**

Another big issue is that in my opinion the discussion section is very difficult to follow in general and very frequently based in statements about processes which has not been measured here. I strongly suggest to the authors avoiding what can be put in their own words in line 316 "this estimation is rather speculative as it only considers …". For instance, all explanations of your results based diatom and cyanobacteria settling are just speculations. You could have measured yourself this rates, if you consider this important in your system. It is part of your hypothesis!

**Answer**: In the manuscript, we aimed to organize the discussion starting from the water column to the benthic processes (first and second chapters). The third chapter describes water and sediment interacting processes, while the last chapter summarizes the most important findings. We tried to re-organize the discussion and hope it flows better in a revised manuscript. The speculative statements were removed from the main text. Regarding the diatom sedimentation, this statement is based on our previous finding in Zilius et al. (2018) and is better integrated, starting from the Introduction and ending with the Discussion. Since the resubmission system does not at the moment allow the submission of a revised version of the manuscript, here, we provide some corrected parts of the Discussion, including a short introduction:

**4 Discussion**

*Using integrated measurements for multiple $NO_3^-$ cycling processes, we assessed pelagic and benthic community activity in relation to $NO_3^-$ availability and other regulatory factors during two contrasting seasons. Combining evidence from process measurements, quantified loads and in situ observations, we also discussed how co-occurring pelagic and benthic processes are important in $NO_3^-$ attenuation, which affects the "coastal filter" function. While we focus only on two-time snapshots, they are most important when pelagic and benthic processes dominate. Such analysis facilitates a mechanistic understanding of the intricate interplay between biological and biogeochemical processes and ecosystem function and services.*

**4.1 The temporal patterns in pelagic $NO_3^-$ uptake**

*In northern latitudes, N availability, primarily driven by $NO_3^-$, fluctuates seasonally, with excess in spring and limitation in summer. Our results show that such $NO_3^-$ dynamics evoke a seasonal succession of phytoplankton from diatom-dominated in spring to cyanobacteria-dominated in summer. These seasonal patterns are also accompanied by increased phytoplankton and bacteria biomass in summer (Zilius et al., 2021). Consequently, changes in the phytoplankton community and biomass profoundly affect ecosystem functioning, including nutrient N fluxes. We found that $NO_3^-$ uptake rates were 65–86% higher in summer than in spring despite low DIN concentrations in the lagoon. Though cyanobacteria prefer $NH_4^+$ (Chaffin and Bridgeman, 2013), they may possess a nutritional strategy to utilize $NO_3^-$ efficiently under N-limiting conditions. While diatoms have a higher affinity for $NO_3^-$, and its excess should stimulate uptake and fast growth (Lomas and Glibert, 2000; Berg et al., 2003; Olofsson et al., 2018), our results show that lower diatom biomass resulted in lower uptake rates in spring. A comparison of $NO_3^-$ uptake rates with estimated phytoplankton N demand (from C fixation measurements) shows that the contribution of $NO_3^-$ to the estimated phytoplankton N demand was relatively small in both spring and summer (~22% on average across sites). This suggests that $NH_4^+$ or DON assimilation or dinitrogen ($N_2$) fixation satisfied most nutritional needs for the growth of phytoplankton. However, this estimation is somewhat underestimated as it only considers the euphotic layer and estimates total autotrophic demand, while in summer, most processes occur under dark conditions.*

*While pelagic $NO_3^-$ uptake in spring is replenished by riverine inputs, uptake in summer would rapidly (~3 hours) deplete the small standing pool in the water column without continuous replete via organic matter mineralization and nitrification. These cycling pathways in the water column are likely supported by the abundance of cyanobacteria. Unlike fast-sinking diatoms, buoyant cyanobacteria remain suspended in the water column. This allows them to eventually contribute to N recycling, which involves processes such as mineralization and nitrification (Peng et al., 2017; Hampel et al., 2019; Chen et al., 2020). We estimated that nitrification in the water column may contribute approximately 11 mmol $NO_3^-$ $m^{-2}$ $d^{-1}$ taking into account the average water column depth of 3.5 m. Based on the fact that the upscaled net $NO_3^-$ uptake in summer was around 9.7 mmol N $m^{-2}$ $d^{-1}$ (Fig. 7) and low concentration prevailed, fast $NO_3^-$ turnover processes likely occurred. We estimated that roughly one-third of the regenerated $NH_4^+$ (~30 mmol N $m^{-2}$ $d^{-1}$ as reported by Zilius et al., 2018)*

*has the potential to undergo oxidation during nitrification. A higher concentration of DON compared to DIN during summer suggests that organic N could also potentially play an important maintaining N cycling in the water column through phytoplankton release, reminilazation to $NH_4^+$ and oxidation or uptake (Wanicke et al., 2009; Korth et al., 2011; Wood and Bukaveckas, 2014; Zilius et al., 2018; Klawonn et al., 2019).*

…………………………

…………………………

**4.3 Summer blooms shadow lagoon, with contrasting effects on different $NO_3^-$ cycling pathways**

*Throughout the year, the turbidity in the water column of the lagoon changes, which impacts the depth of the euphotic zone (Zilius et al., 2014). Light scattering is primarily caused by buoyant cyanobacteria that reduce the euphotic zone to ~1 m in summer. As a result, self-shaded cyanobacteria and other microorganisms experience periods of darkness. However, through the combined effects of gas vesicles and wind-driven vertical mixing, cyanobacteria and other microorganisms remain in continuous motion, temporarily accessing the narrow euphotic zone where they gain energy for their $NO_3^-$ uptake and metabolism. Based on our previous metagenome analyses in the lagoon, the presence in the summer of the marker genes responsible for assimilative $NO_3^-$ reduction in surface and bottom waters was mainly attributed to Cyanobacteria, followed by Firmicutes and Euryarcheota (Broman et al., 2021). This implies that the cyanobacterial community can assimilate $NO_3^-$ throughout the entire water column, even below the euphotic zone, and thus competing with heterotrophic microorganisms (Chaffin and Bridgeman, 2014; Hampel et al., 2019). The results show that the rates of $NO_3^-$ uptake in light were consistently higher than those in darkness (by 16–62%), indicating more effective cyanobacteria uptake during the daylight hours within the euphotic zone. In the summer, the dominant Planktotrix genus is adapted to lower light intensities than other cyanobacteria, making it even more competitive (Post et al., 1985). Regarding biomass, cyanobacteria dominate over bacteria, and thus are a key component in N cycling pathways (Broman et al., 2021; Zilius et al., 2018, 2021).*

…………………………

…………………………

**4.4 Importance of pelagic and benthic $NO_3^-$ cycling in the context of the lagoon's N budget**

*Simultaneous measurements of pelagic and benthic $NO_3^-$ processes allowed us to depict cycling and its contribution to $NO_3^-$ retention during two contrasting seasons (Fig. 7). During the spring season, the average pelagic assimilation of $NO_3^-$ (1.2 mmol N $m^{-2}$ $d^{-1}$), derived from the upscaled site-specific measurements accounting for the northern (45%) and south-central areas (55%) (Zilius et al., 2018), was estimated to equal to 18% of average daily N load (6.6 mmol N $m^{-2}$ $d^{-1}$ in March–May) delivered to the lagoon. Upscaled net daily $NO_3^-$ uptake by sediments (2.7 mmol N $m^{-2}$ $d^{-1}$) was, on average, equivalent to 41% of the N load. Both pelagic and benthic processes accounted for the same retention when comparing inputs and outputs to the system (60% of load retention) conducted by Vybernaite-Lubiene et al. (2017). In summer, pelagic demand for $NO_3^-$ increased and exceeded by 85% riverine inputs to the lagoon (1.5 mmol N $m^{-2}$ $d^{-1}$). Higher $NO_3^-$ uptake in the water column than in sediment, particularly in the south-central area, was due to higher cyanobacterial biomass, which may reflect longer water renewal time in comparison to the northern site (Zilius et al., 2014; Vaičiūte et al., 2021). Overall, $NO_3^-$ assimilation, especially in summer when N limitation exists, represents an important N cycling pathway in the water column after $NH_4^+$ assimilation (Broman et al., 2021). This demand for $NO_3^-$ far exceeds N inputs via biological dinitrogen fixation (2.1 mmol N $m^{-2}$ $d^{-1}$; Zilius et al., 2021) as well as $NO_3^-$ release from sediments and river inputs.*

*However, it remains questionable whether nitrification in the water column can sustain such NO$_3^-$ demand. Therefore, future studies should also include measurements of NO$_3^-$ production via nitrification and other cycling pathways (Fig. 7).*

Specific comments

Line 118. The uptake experiments with the pelagic samples were done in an outdoor tank and no information is provided about what irradiance was used, only it is mentioned that incubations were shaded to prevent high light exposure. Since N-demand in pelagic communities is closely connected with photosynthesis rate, information about irradiance should be more specific. Was this irradiance measured? Since they were outdoor incubations, irradiance during incubations was not constant.

**Answer:** Yes, the pelagic rates were measured under natural irradiance in an outdoor tank following our previous experimental approach by Broman et al. (2021) and Zilius et al. (2021). Information about average light conditions was added and now reads "*Water samples were placed in an outdoor tank and incubated at temperatures comparable to those in the lagoon (15.0 °C and 21.0 °C, respectively, in May and August). Irradiance levels within the tanks were 480 (in May) and 880 µmol s$^{-1}$ m$^{-2}$ (in August) which approximated light levels at (mid-depth) in the lagoon*".

- Broman, E., M. Zilius, A. Samuiloviene, I. Vybernaite-Lubiene, T. Politi, I. Klawonn, M. Voss, F.J.A. Nascimento, S. Bonaglia, 2021. Active DNRA and denitrification in oxic hypereutrophic waters. Water Research 194: 116954. DOI:10.1016/j.watres.2021.116954
- Zilius, M., I. Vybernaite-Lubiene, D. Vaiciute, D. Overlingė, E. Grinienė, A. Zaiko, S. Bonaglia, I. Liskow, M. Voss, A. Andersson, S. Brugel, T. Politi, P.A. Bukaveckas, 2021. Spatiotemporal patterns of N$_2$ fixation in coastal waters derived from rate measurements and remote sensing. Biogeosciences 18: 1857–1871. DOI:10.5194/bg-18-1857-2021

Line 135. Benthic NO$_3^-$ fluxes were measured in a temperature-controlled room instead of in outdoor tanks, why? The cores collected from the shallower site were incubated at a constant irradiance. Why this was not done for incubations with the pelagic samples?

**Answer**: Benthic NO$_3^-$ fluxes and dissimilative reduction pathways (denitrification and DNRA) were measured in a climate-controlled room to maintain more stable conditions, particularly temperature, as the final product of denitrification is gas, which dissolution is temperature-dependent. Simulating low-light conditions (1 of 4 incubations) or darkness (3 of 4 incubations) was more accessible in a climate-controlled room where direct sunlight could be avoided and low irradiance (~60 µmol s$^{-1}$ m$^{-2}$) produced to surface sediment.

For pelagic uptake rates, we chose to incubate under natural light conditions, as higher irradiance (879 µmol s$^{-1}$ m$^{-2}$ in summer) could not be simulated indoors.

We acknowledge that variation in irradiance during the daytime can change, especially in summer. Still, it is also important to note that incubations lasted very short, from 1.5 hours in summer to 4 hours in spring. As a result, we gain less of the effect of light variability during the incubations.

L134 and L156. Potential readers would appreciate being more specific in these heading subsections. What processes measurements? What other methods?

**Answer**: We have specified titles of subsections:

2.3 Measurements of benthic NO$_3^-$ flux and dissimilatory reduction processes
2.4 Analytical methods and phytoplankton analysis
L165. How did you calculate biomass from biovolume? This is not mention in the reference you cited. You should provide a brief description of the conversion of phytoplankton counts to biomass since this is not straightforward at all. Another matter is why there is not replicate analysis of these

samples? You are making a lot of interpretations of your results based on the differences in the phytoplankton community from just two no replicated samples.

**Answer**: According to Olenina et al. (2006), cell numbers were multiplied by species-specific mean cell volumes to obtain biovolume, assuming a density of 1 g mL$^{-1}$. This is a primary reference used in the Baltic region and approved by HELCOM (see other references supporting our choice for Olenina et al. 2026). Missing information was added:

*"Phytoplankton community composition was determined using the Utermöhl method (Utermöhl, 1958) according to HELCOM recommendations (HELCOM, 2017). Phytoplankton biomass (mg L$^{-1}$) was calculated according to Olenina et al. (2006) whereby cell numbers were multiplied by species-specific mean cell volumes, assuming a density of 1 g mL$^{-1}$."*

We agree that more than one replicate would be helpful in the work. However, microscopic counting is frequently based on single-sample analysis, mainly due to the immense effort of time needed to identify phytoplankton species and count their cells. Therefore, as in most other studies, we applied this approach here (see studies below). Last, our goal was simply to confirm diatom vs. cyanobacteria-dominated communities during the two sampling periods.

- Zakrisson, A., Larsson, U. and glander, H. 2014. Do Baltic Sea diazotrophic cyanobacteria take up combined nitrogen in situ? J. Plankton Res. 36(5): 1368–1380. doi:10.1093/plankt/fbu053
- Olofsson, M., Klawonn, I. and Karlson, B. 2021. Nitrogen fixation estimates for the Baltic Sea indicate high rates for the previously overlooked Bothnian Sea. Ambio 50, 203–214. doi:10.1007/s13280-020-01331-x
- Sörenson, E., Farnelid, H., Lindehoff, E. and Legrand, C. 2020. Resource partitioning between phytoplankton and bacteria in the Coastal Baltic Sea. Front. Mar. Sci. 7: 608244. doi: 10.3389/fmars.2020.608244
- Klawonn, I., Bonaglia, S., Whitehouse, M.J. et al. 2019. Untangling hidden nutrient dynamics: rapid ammonium cycling and single-cell ammonium assimilation in marine plankton communities. ISME J. 13, 1960–1974. doi:10.1038/s41396-019-0386-z
- Wasmund, N., Tuimala, J., Suikkanen, S., Vandepitte, L. and Kraberg, A. 2011. Long-term trends in phytoplankton composition in the western and central Baltic Sea. J. Marine Sys. 87(2): 145–159. doi:10.1016/j.jmarsys.2011.03.010.

L182. You set a significance level in the statistical method section, why sometime you use it and other times you give the exact value of p? You must be consistent!

**Answer**: it was corrected.

L205. I do not understand this: "… were averaged based on surface and bottom water measurements."

**Answer**: This is our mistake in table captions, as samples were column integrated (see method section), as written in the methods: "……… *vertically integrated water samples that represent the whole water column were collected with a Niskin bottle and transferred*...". The corrected caption is:

*Table 1. Measured in situ environmental variables. kd – extinction coefficient of the light in the water column. The values show the mean ± standard error based on three replicates except for the central site.*

L254. Please, make clear that you used the C:N ratio as well to calculate N-demand form C-fixation.

**Answer**: We have reformulated the sentence as suggested "*Rates of C-fixation during the daytime were used to estimate pelagic community N demand based on the C:N ratio in the biomass.*"

Figures and Tables

Fig. 1 and legend. There is no blue circle in my pdf copy

**Answer:** The reviewer is right; we have correct figure captions. Following the comment of reviewer, we have also added an additional panel indicating lagoon outflow and the exchange site with the sea.

[Figure]

.

*Figure 1 Captions. Satellite image by OLI/Landsat-8 (18/09/2014) showing summer blooms in the Curonian Lagoon with the sampling sites (stars) representing the northern and south-central regions and monitoring site at the Nemunas River (black circle). The black line indicates a state border between two countries. The outflow is at the northern end of the lagoon near Klaipeda.*

Fig. 6. You need to include legends on this figure.

**Answer**: We added a legend, which was missing due to a technical issue when the file was exported from the drawing program.

[Figure]

*Figure 6. Net daily NO$_3^-$ fluxes (A,D), total denitrification (D$_{tot}$; B,E), denitrification fuelled by water column NO$_3^-$ (D$_w$) and coupled to nitrification (D$_n$), and total dissimilative nitrate reduction to ammonium (DNRA$_{tot}$; C,F), DNRA fueled by water column NO$_3^-$ (DNRA$_w$) and coupled to nitrification (DNRA$_n$) measured at the two sites in the Curonian Lagoon in spring (May) and summer (August) 2021. The positive and negative values of NO$_3^-$ fluxes indicate the release and uptake of nutrients, respectively. Data shown are mean values and standard errors (n=5).*

Table 2. What are the differences between data presented in figure 6 and those of table 2?

**Answer**: Thank you for the comment. Figure 6 contains net daily fluxes and dissimilatory reduction process rates (see captions above) derived by multiplying the hourly rates by the length of the day and night. Table 2 (see captions and table below) contains hourly dissimilatory reduction rates, which are needed to show the difference between light and dark, as irradiance affects rates.

*Table 2. Hourly rates (μmol N m$^{-2}$ h$^{-1}$) of benthic dissimilatory nitrate (NO$_3^-$) reduction through denitrification and dissimilative nitrate reduction to ammonium (DNRA) under different light conditions in spring at northern (sandy) and central (muddy) sites in the Curonian Lagoon. D$_{tot}$ – total denitrification, D$_w$ – denitrification fueled by NO$_3^-$ from the overlaying water column, D$_n$ – coupled nitrification-denitrification, DNRA$_w$ – DNRA fueled by NO$_3^-$ from the overlaying water column, DNRA$_n$ – coupled nitrification-DNRA. Data shown are mean values and standard errors (n=5).*

| NO$_3^-$ reduction process | | Central site | | Northern site | | |
|---|---|---|---|---|---|---|
| | | Spring | Summer | Spring | | Summer |
| | | Dark | Dark | Light | Dark | Dark |
| Denitrification | D$_{tot}$ | 141.0±5.4 | 48.4±2.6 | 24.8±4.1 | 77.0±15.3 | 6.7±0.9 |
| | D$_w$ | 101.9±2.4 | 1.3±0.6 | 22.5±6.4 | 65.0±1.0 | 0.1±0.0 |
| | D$_n$ | 39.1±1.7 | 47.2±2.8 | 2.4±3.0 | 12.0±2.9 | 6.6±0.9 |
| DNRA | DNRA$_{tot}$ | 12.7±1.0 | 21.9±1.6 | 14.9±3.8 | 1.8±0.1 | 2.3±0.3 |
| | DNRA$_w$ | 9.2±0.8 | 0.6±0.3 | 13.2±3.9 | 1.5±0.2 | <0.05 |
| | DNRA$_n$ | 3.5±0.2 | 21.3±1.5 | 1.7±1.4 | 0.3±0.0 | 2.3±0.3 |